# Consensus-Related Performance of Triplex MASs Based on Partial Complete Graph Structure

**DOI:** 10.3390/e24091296

**Published:** 2022-09-14

**Authors:** Jicheng Bian, Da Huang, Jiabo Xu, Zhiyong Yu

**Affiliations:** 1School of Mathematics and Physics, Xinjiang Institute of Engineering, Urumqi 830023, China; 2Key Laboratory of New Energy and Materials Research, Xinjiang Institute of Engineering, Urumqi 830023, China; 3School of Information Engineering, Xinjiang Institute of Engineering, Urumqi 830023, China; 4College of Mathematics and System Science, Xinjiang University, Urumqi 830017, China

**Keywords:** multi-agent system, consensus, coherence, Cartesian product, Laplacian spectrum

## Abstract

This article mainly studies first-order coherence related to the robustness of the triplex MASs consensus models with partial complete graph structures; the performance index is studied through algebraic graph theory. The topologies of the novel triplex networks are generated by graph operations and the approach of graph spectra is applied to calculate the first-order network coherence. The coherence asymptotic behaviours of the three cases of the partial complete structures are analysed and compared. We find that under the condition that the number of nodes in partial complete substructures *n* tends to infinity, the coherence asymptotic behaviour of the two sorts of non-isomorphic three-layered networks will be increased by $\frac{r-1}{2(r+1)}$, which is irrelevant to the peripheral vertices number *p*; when *p* tends to infinity, adding star copies to the original triplex topologies will reverse the original size relationship of the coherence under consideration of the triplex networks. Finally, the coherence of the three-layered networks with the same sorts of parameters, but non-isomorphic graphs, are simulated to verify the results.

## 1. Introduction

The consensus problem is a significant research field of multi-agent systems(MASs); it requires that all nodes in the networked system reach a common physical state value based on some control protocols. The nodes are designed to cooperate effectively under the topologies of the system in order to reach the predetermined aims.

As a significant interdisciplinary field, consensus-related research has received much attention from both engineers and scholars in the past two decades, and there exist many applications or research works on the consensus problem, for instances, formation control, sensor networks, decision making, and so on. Scholars have conducted significant valuable research on consensus from various aspects, such as the order of system (first or second order [1,2,3,4,5,6,7,8,9,10,11,12,13,14,15,16,17,18,19]), ways of communication (continuous or intermittent [4,7], control methods (adaptive control [5], intermittent control [7], sliding mode control [19]), convergence time (finite time or fixed time [6,8]) and so on.

For solving consensus problems, the connecting relations among nodes are characterised by the graph of the network, and the convergence index of the dynamics model, such as consensus speed [1,9] and network coherence [11,12,13,14,15,16], can be determined by the second smallest Laplacian eigenvalue $\lambda_{2}$ and the Laplacian spectrum. Similarly, as a branch of coordination problems, synchronization problems are always connected with the notion of system topologies [20,21,22,23,24,25].

There exists significant coordination-related research on the Laplacian eigenvalues [1,9,10,11,12,13,14,15,16,17,18,19,20,21,22] of the network graph. In these enlightening works, Ref. [1] has showed that the algebraic connectivity $\lambda_{2}$ of the system topology can characterise the speed of the convergence process of the system. Ref. [10] has investigated how the Laplacian eigenvalues determines the robustness, and derived the consensus speed and the $H_{2}$ norms of some classic network structures.

The concept of network coherence has been proposed in [11,12], and the fact that the robustness-related coherence can be characterised by the nonzero Laplacian eigenvalues was obtained. In Ref. [14], an analytical expression for the leader–follower coherence is determined depending on the number of leader nodes and network parameters. In [15], based on the tree construction, the authors show that the consensus in symmetric and asymmetric trees becomes better with an increasing number of leader nodes.

In recent years, multilayered structure has become a hot research topic of complex networks [26,27], and it has many real examples of application, such as interactions between the internet and power grid, and aviation and transportation networks.

Since many real-world networks have multi-layered topologies, it is natural and meaningful to extend the application of the Laplacian spectrum for consensus theory to multi-layered topologies. From the view of application, the star-shaped structure is a sort of classic computer network graph, and it is widely studied in many fields, including the coordination-related problem [10,16,17,21,22,24,28]. It is natural to consider the consensus problem with both multi-layered structure and star subgraphs.

Combined with the above description, this paper considers several partial complete graph-based triplex networks constructed by the graph operations, and this work mainly focuses on the first-order robustness of consensus to communication noise under the triplex systems. The main novelties of this paper are listed as follows:1.Several novel triplex networks with partial complete graph structures are constructed by graph operations, and the related three-layered networks are also proposed;2.Theory on graph spectra is applied to obtain the Laplacian spectrums. Methods on analysis are applied for the calculation of network coherence, and new results on the asymptotic behaviour are acquired;3.We find that, under the condition that the number of nodes in partial complete substructures *n* tends to infinity, the coherence asymptotic behaviour of the considered non-isomorphic three-layered networks will increased by $\frac{r-1}{2(r+1)}$, which is irrelevant to the peripheral vertices parameter *p*. When *p* tends to infinity, the adding star copies operation will reverse the original size relationship of the coherence under the consideration of the novel triplex networks.

This article is organised as follows. In Section 2, notations on graph theory are summarised and the relations between performance and Laplacian eigenvalues are described. In Section 3, the topology constructions of the triplex systems and main results are given. In Section 4, the simulation results are compared and analysed.

## 2. Preliminaries

### 2.1. Graph Theory and Notations

A complete graph of *n* vertices is denoted by $K_{n}$, and a star graph with k−1 leaves is denoted by $S_{k}$. The path with *m* vertices is denoted by $P_{m}$, and the fan graph with *p* vertices is denoted by $F_{p}$. The wheel graph with *r* vertices is denoted as $W_{r}$. $E_{k}$ is defined to be the empty graph with k vertices, where empty graph means the graph has no edges between any pair of vertices. Let *G* be a graph with vertex set $V=\{v_{1},v_{2},\cdots ,v_{N}\}$, and its edge set is defined by $E=\{\left(v_{i},v_{j}\right)|i,j=1,2,\cdots ,N;i\neq j\}$. The adjacency matrix of *G* is defined as $A\left(G\right)={\left[a_{ij}\right]}_{N}$, where $a_{ij}$ is the weight of the edge $\left(v_{i},v_{j}\right)$. To an undirected graph, $a_{ij}=a_{ji}$. All the edges in the undirected networks of this paper are 0–1 weighted, that is, $a_{ij}=\left\{\begin{array}{cc} 1, & (v_{i},v_{j})\in E;\\ 0, & (v_{i},v_{j})\notin E. \end{array}\right.$. The Laplacian matrix of *G* is defined as L(G)=D(G)−A(G), where D(G) is the diagonal degree matrix of *G* defined by $D\left(G\right)=diag\left(d_{1},d_{2},\ldots ,d_{N}\right)$ with $d_{i}=\sum_{j\neq i}a_{ij}$. The Laplacian spectrum is denoted by: $SL\left(G\right)=\left(\begin{array}{cccc} \lambda_{1}\left(G\right) & \lambda_{2}\left(G\right) & \cdots & \lambda_{p}\left(G\right)\\ l_{1} & l_{2} & \cdots & l_{p} \end{array}\right)$, where $\lambda_{1}\left(G\right)<\lambda_{2}\left(G\right)<\cdots <\lambda_{p}\left(G\right)$ are the eigenvalues of L(G), and $l_{1},l_{2},\ldots ,l_{p}$ are the multiplicities of the eigenvalues [29].

Denote the corona of two graphs by ‘∘’ [30,31], the join operation of two graphs by ‘▿’ [29,32], and the Cartesian product of two graphs by ‘×’ [32,33,34].

### 2.2. Relations for the Coherence and Laplacian Eigenvalues

(Refs. [11,12,13,14,15]) The first-order system with disturbance is
(1)$$\begin{array}{c} \dot{x}\left(t\right)=-L\left(G\right)x\left(t\right)+\vartheta \left(t\right) \end{array}$$
with $x\left(t\right)\in R^{N}$ and where $\vartheta \left(t\right)\in R^{N}$ is a vector of uncorrelated noise. L(G) is the Laplacian matrix.

**Definition** **1**([11,12]). *The first-order network coherence is defined as the mean steady-state variance:*
$$\begin{array}{c} H_{f}=\lim_{t\to \infty }\frac{1}{N}\sum_{i=1}^{N}\mathrm{Var}\left\{x_{i}\left(t\right)-\frac{1}{N}\sum_{j=1}^{N}x_{j}\left(t\right)\right\}. \end{array}$$

It has been proved that the first-order coherence $H_{f}$ is determined by the spectrum of *L*. Let the L− eigenvalues be $0=\lambda_{1}<\lambda_{2}\leq \ldots \leq \lambda_{N}$; then, the first-order coherence can be characterised by
(2)$$\begin{array}{c} H_{f}=\frac{1}{2N}\sum_{i=2}^{N}\frac{1}{\lambda_{i}}. \end{array}$$

## 3. Main Results

The layered partial complete networks of this paper have topologies composed by linking the counterpart nodes of the different layers. The following subsections propose the triplex partial complete networks and calculate the coherence.

### 3.1. The Coherence for the Triplex Networks with $G_{i}\left(n,p\right)$

In this subsection, three sorts of triplex partial complete subgraph structures $G_{i}\left(n,p\right)$, i=1,2,3 and the corresponding extended graph $J_{i}\left(n,p,r\right),i=1,2$ are considered.

Case 1: As shown in Figure 1, the complete subgraph structure in the middle composed by blue nodes is denoted by $K_{n}$; then, the partial complete graph can be denoted as $G_{1}\left(n,p\right):=\left(K_{n}▿E_{p}\right)\times P_{3}$, and the corresponding network with dynamics (1) by $G_{1}$.

Then, it can be derived that
$$SL\left[G_{1}\left(n,p\right)\right]=\left(\begin{array}{ccccccccc} 0 & n+p & n & 1 & 1+n+p & 1+n & 3 & 3+n+p & 3+n\\ 1 & n & p-1 & 1 & n & p-1 & 1 & n & p-1 \end{array}\right),$$

Hence, the first-order coherence of $G_{1}$ is
$$H\left(G_{1}\right)=\frac{1}{6(n+p)}\left(\frac{n}{\left(n+p\right)}+\frac{p-1}{n}+1+\frac{n}{1+n+p}+\frac{p-1}{1+n}+\frac{1}{3}+\frac{n}{3+n+p}+\frac{p-1}{3+n}\right),$$

(i).If p→∞, then it can be obtained that $H\left(G_{1}\right)\to \frac{1}{6n}+\frac{1}{6(1+n)}+\frac{1}{6(3+n)}$;(ii).If n→∞, then $H(G_{1})\to 0$.

Define $J_{1}\left(n,p,r\right):=\left[\left(K_{n}▿E_{p}\right)\times P_{3}\right]\circ E_{r}$; this is abbreviated as $J_{1}$. Let it be the topology (see Figure 2) of the corresponding noisy network $J_{1}$; then, $SL(J_{1})$ has the following description:(1).$0,r+1\in SL(J_{1})$ repeated only once’(2).$\frac{\left(n+p+r+1\right)\pm \sqrt{{\left(n+p+r+1\right)}^{2}-4\left(n+p\right)}}{2}\in SL\left(J_{1}\right)$ repeated *n* times;(3).$\frac{\left(n+r+1\right)\pm \sqrt{{\left(n+r+1\right)}^{2}-4n}}{2}\in SL\left(J_{1}\right)$ repeated p−1 times;(4).$\frac{\left(r+2\right)\pm \sqrt{{\left(r+2\right)}^{2}-4}}{2}\in SL\left(J_{1}\right)$ with multiplicity 1;(5).$\frac{\left(n+p+r+2\right)\pm \sqrt{{\left(n+p+r+2\right)}^{2}-4\left(n+p+1\right)}}{2}\in SL\left(J_{1}\right)$ repeated *n* times;(6).$\frac{\left(n+r+2\right)\pm \sqrt{{\left(n+r+2\right)}^{2}-4\left(n+1\right)}}{2}\in SL\left(J_{1}\right)$ repeated p−1 times;(7).$\frac{\left(r+4\right)\pm \sqrt{{\left(r+4\right)}^{2}-12}}{2}\in SL\left(J_{1}\right)$ repeated 1 time;(8).$\frac{\left(n+p+r+4\right)\pm \sqrt{{\left(n+p+r+4\right)}^{2}-4\left(n+p+3\right)}}{2}\in SL\left(J_{1}\right)$ repeated *n* times;(9).$\frac{\left(n+r+4\right)\pm \sqrt{{\left(n+r+4\right)}^{2}-4\left(n+3\right)}}{2}\in SL\left(J_{1}\right)$ repeated p−1 times;(10).$1\in SL(J_{1})$ repeated 3(n+p)(r−1) times.

**Figure 2 entropy-24-01296-f002:**
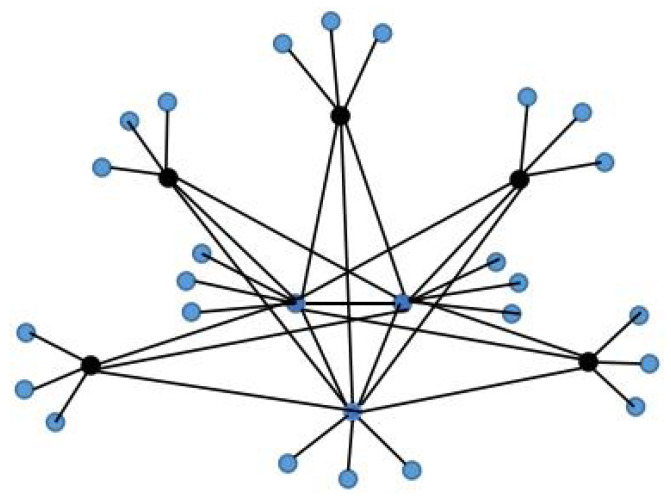
An example of one layer of $J_{1}\left(n,p,r\right)$ with n=3 and p=5, r=3.

Therefore, the first-order coherence $H(J_{1})$ can be derived as:$$\begin{array}{cc} H(J_{1})= & \frac{1}{6(n+p)(1+r)}[\frac{1}{r+1}+\frac{n+p+r+1}{n+p}+\frac{n+r+1}{n}+\left(2+r\right)+\frac{2+n+p+r}{n+p+1}\\ \; & +\frac{n+2+r}{n+1}+\frac{4+r}{3}+\frac{4+n+p+r}{3+n+p}+\frac{4+n+r}{3+n}+3\left(n+p\right)\left(r-1\right)] \end{array}$$

Therefore,

(i).When *n* and *r* are fixed, $H\left(J_{1}\right)\to \frac{r-1}{2(1+r)}$ as p→∞, and it is irrelevant to *n*;(ii).Let n→∞; then, $H\left(J_{1}\right)\to \frac{r-1}{2(1+r)}$, it is irrelevant to *p*;(iii).If r→∞, then

$H\left(J_{1}\right)\to \frac{1}{6(n+p)}\left(\frac{1}{n+p}+\frac{1}{n}+1+\frac{1}{n+p+1}+\frac{1}{n+1}+\frac{1}{3}+\frac{1}{3+n+p}+\frac{1}{3+n}\right)+\frac{1}{2}$.

**Remark** **1.**
*Different from the result (i) of the asymptotic properties of $H(G_{1})$, the limitation of $H(J_{1})$ in result (i) is only relevant to the parameter r as p tends to infinity. In the result (ii) of $H(G_{1})$ and $H(J_{1})$, when n→∞, the coherence limitation increases by $\frac{r-1}{2(r+1)}$ and we have $\lim_{p\to \infty }H\left(J_{1}\right)=\lim_{n\to \infty }H\left(J_{1}\right)$.*


Case 2: The structure composed by the hub nodes is defined the same as Case 1, and if the generalised leaf nodes are connected into a path (see Figure 3), then the triplex graph of the system can be denoted as $G_{2}\left(n,p\right):=\left(K_{n}▿P_{p}\right)\times P_{3}$, and it can be derived that $SL[G_{2}\left(n,p\right)]$ has the description: $0,1,3\in SL[G_{2}\left(n,p\right)]$ with multiplicity 1, respectively; n+p, $n+p+1,\;n+p+3\in SL[G_{2}\left(n,p\right)]$ with multiplicity *n*, respectively; $n+4sin^{2}\left(\frac{k\pi }{2p}\right)\in SL\left[G_{2}\left(n,p\right)\right]$ with multiplicity 1, where k=1,2,⋯,p−1; $n+1+4sin^{2}\left(\frac{k\pi }{2p}\right)\in SL\left[G_{2}\left(n,p\right)\right]]$ with multiplicity 1, where k=1,2,⋯,p−1; $n+3+4sin^{2}\left(\frac{k\pi }{2p}\right)\in SL\left[G_{2}\left(n,p\right)\right]$ with multiplicity 1, where k=1,2,⋯,p−1.

Therefore, the first-order coherence of noisy network $G_{2}$ can be characterized as
$$\begin{array}{cc} H(G_{2})= & \frac{1}{6(n+p)}(1+\frac{1}{3}+\frac{n}{n+p}+\frac{n}{n+p+1}+\frac{n}{n+p+3}+\sum_{k=1}^{p-1}\frac{1}{n+4sin^{2}\left(\frac{k\pi }{2p}\right)}\\ \; & +\sum_{k=1}^{p-1}\frac{1}{n+1+4sin^{2}\left(\frac{k\pi }{2p}\right)}+\sum_{k=1}^{p-1}\frac{1}{n+3+4sin^{2}\left(\frac{k\pi }{2p}\right)}) \end{array}$$

(i).Let *n* be fixed and p→∞; then, it can be obtained that:
$$\begin{array}{c} \lim_{p\to \infty }H\left(G_{2}\right)=\frac{1}{6}\left(\frac{1}{\sqrt{n(n+4)}}+\frac{1}{\sqrt{\left(n+5)(n+1\right)}}+\frac{1}{\sqrt{\left(n+7)(n+3\right)}}\right) \end{array}$$(ii).If *p* is fixed, let n→∞; then, $H(G_{2})\to 0$;

Define $J_{2}\left(n,p,r\right):=\left[\left(K_{n}▿P_{p}\right)\times P_{3}\right]\circ E_{r}$, abbreviate it as $J_{2}$, and let it be the topology (see Figure 4) of noisy network $J_{2}$; then, $SL(J_{2})$ has the following description:

(1). 0 and $r+1\in SL(J_{2})$ with multiplicity 1;

(2). $\frac{\left(2+r\right)\pm \sqrt{{\left(2+r\right)}^{2}-4}}{2}\in SL\left(J_{2}\right)$ with multiplicity 1;

(3). $\frac{\left(4+r\right)\pm \sqrt{{\left(4+r\right)}^{2}-12}}{2}\in SL\left(J_{2}\right)$ with multiplicity 1;

(4). $\frac{\left(n+p+r+1\right)\pm \sqrt{{\left(n+p+r+1\right)}^{2}-4\left(n+p\right)}}{2}\in SL\left(J_{2}\right)$ repeated *n* times;

(5). $\frac{\left(n+p+r+2\right)\pm \sqrt{{\left(n+p+r+2\right)}^{2}-4\left(n+p+1\right)}}{2}\in SL\left(J_{2}\right)$ repeated *n* times;

(6). $\frac{\left(n+p+r+4\right)\pm \sqrt{{\left(n+p+r+4\right)}^{2}-4\left(n+p+3\right)}}{2}\in SL\left(J_{2}\right)$ repeated *n* times;

(7). $\frac{\left(n+4sin^{2}\left(\frac{k\pi }{2p}\right)+r+1\right)\pm \sqrt{{\left(n+4sin^{2}\left(\frac{k\pi }{2p}\right)+r+1\right)}^{2}-4\left(n+4sin^{2}\left(\frac{k\pi }{2p}\right)\right)}}{2}\in SL\left(J_{2}\right)$ with multiplicity 1, k=1,2,⋯,p−1;

(8). $\frac{\left(n+4sin^{2}\left(\frac{k\pi }{2p}\right)+r+2\right)\pm \sqrt{{\left(n+4sin^{2}\left(\frac{k\pi }{2p}\right)+r+2\right)}^{2}-4\left(n+4sin^{2}\left(\frac{k\pi }{2p}\right)+1\right)}}{2}\in SL\left(J_{2}\right)$ with multiplicity 1, k=1,2,⋯,p−1;

(9). $\frac{\left(n+4sin^{2}\left(\frac{k\pi }{2p}\right)+r+4\right)\pm \sqrt{{\left(n+4sin^{2}\left(\frac{k\pi }{2p}\right)+r+4\right)}^{2}-4\left(n+4sin^{2}\left(\frac{k\pi }{2p}\right)+3\right)}}{2}\in SL\left(J_{2}\right)$ with multiplicity 1, k=1,2,⋯,p−1;

(10). $1\in SL(J_{2})$ repeated 3(n+p)(r−1) times.

**Figure 4 entropy-24-01296-f004:**
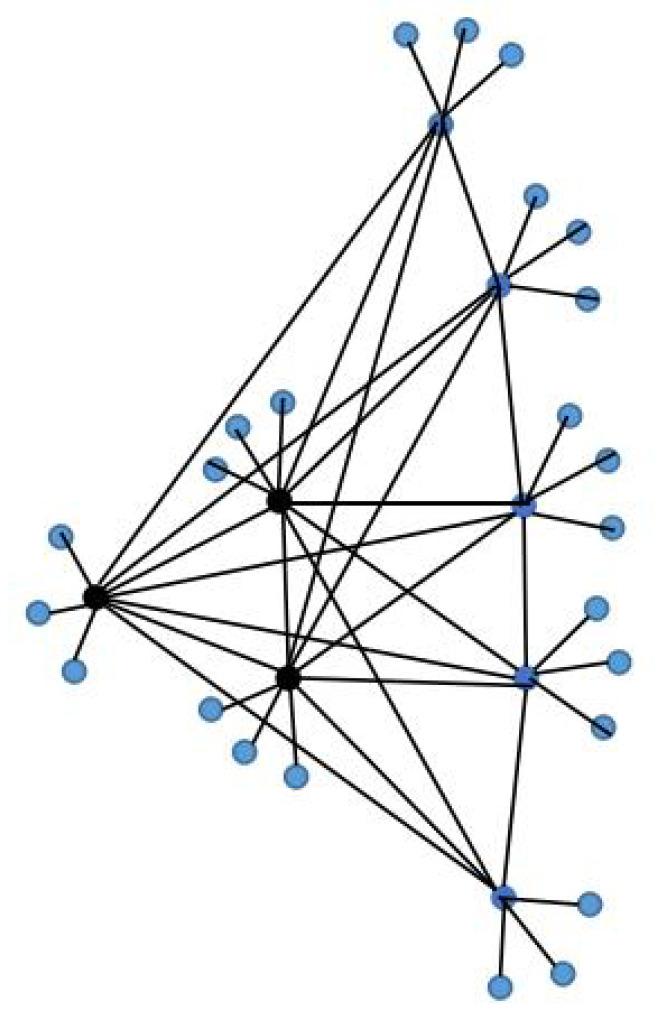
An example of one layer of $J_{2}\left(n,p,r\right)$ with n=3 and p=5, r=3.

Therefore, the coherence $H(J_{2})$ can be calculated as:$$\begin{array}{cc} H(J_{2})= & \frac{1}{6(n+p)(1+r)}(\frac{1}{r+1}+\left(2+r\right)+\frac{4+r}{3}+\frac{n+p+1+r}{n+p}+\frac{n+p+2+r}{n+p+1}\\ \; & +\frac{n+p+4+r}{n+p+3}+3\left(p-1\right)+\left(1+r\right)\sum_{k=1}^{p-1}\frac{1}{n+4sin^{2}\frac{k\pi }{2p}}\\ \; & +\left(1+r\right)\sum_{k=1}^{p-1}\frac{1}{n+4sin^{2}\frac{k\pi }{2p}+1}+\left(1+r\right)\sum_{k=1}^{p-1}\frac{1}{n+4sin^{2}\frac{k\pi }{2p}+3}\\ \; & +3(n+p)(r-1)), \end{array}$$

Hence, we have

(i).When *n* and *r* are fixed,
$$\begin{array}{cc} \lim_{p\to \infty }H\left(J_{2}\right)= & \frac{1}{6}\left(\int_{0}^{1}\frac{1}{n+4sin^{2}\frac{\pi x}{2}}dx+\int_{0}^{1}\frac{1}{n+4sin^{2}\frac{\pi x}{2}+1}dx+\int_{0}^{1}\frac{1}{n+4sin^{2}\frac{\pi x}{2}+3}dx\right)\\ \; & +\frac{r}{2(1+r)}=\lim_{p\to \infty }H\left(G_{2}\right)+\frac{r}{2(1+r)}; \end{array}$$(ii).Let n→∞, $H\left(J_{2}\right)\to \frac{r-1}{2(1+r)}$ holds;(iii).If r→∞,
$$\begin{array}{cc} H(J_{2})\to & \frac{1}{6(n+p)}(\frac{4}{3}+\frac{1}{n+p}+\frac{1}{n+p+1}+\frac{1}{n+p+3}+\sum_{k=1}^{p-1}\frac{1}{n+4sin^{2}\frac{k\pi }{2p}}\\ \; & +\sum_{k=1}^{p-1}\frac{1}{n+4sin^{2}\frac{k\pi }{2p}+1}+\sum_{k=1}^{p-1}\frac{1}{n+4sin^{2}\frac{k\pi }{2p}+3})+\frac{1}{2}. \end{array}$$

**Remark** **2.**
*By comparing the pairs of results on $H\left(G_{1}\right),H\left(J_{1}\right)$ and $H\left(G_{2}\right),H\left(J_{2}\right)$, one can see that the coherence asymptotic behaviour of the above cases (ii) increases by $\frac{r-1}{2(1+r)}$, and we have $\lim_{p\to \infty }H\left(J_{1}\right)=\lim_{n\to \infty }H\left(J_{1}\right)=\lim_{n\to \infty }H\left(J_{2}\right)$; in addition, for subcase (i), $\lim_{p\to \infty }H\left(J_{2}\right)>\lim_{p\to \infty }H\left(J_{1}\right)$. However, in the previous subsection, we have $\lim_{p\to \infty }H\left(G_{2}\right)<\lim_{p\to \infty }H\left(G_{1}\right)$, which means, under the triplex structures, the adding of the star-subgraph copies can change the size relation of the first-order coherence as p→∞. To the subcase (iii), it can be obtained that, when p≥4,n≥3, $\lim_{r\to \infty }H\left(J_{2}\right)>\lim_{r\to \infty }H\left(J_{1}\right)$.*


Case 3: Similarly, define the network $G_{3}$ with the triplex graph $G_{3}\left(n,p\right):=\left[\left(K_{n}▿C_{p}\right)\times P_{3}\right]$. Then, it can be derived that
$$\begin{array}{cc} H(G_{3})= & \frac{1}{6(n+p)}(1+\frac{1}{3}+\frac{n}{n+p}+\frac{n}{n+p+1}+\frac{n}{n+p+3}+\sum_{k=1}^{p-1}\frac{1}{n+4sin^{2}\left(\frac{k\pi }{p}\right)}\\ \; & +\sum_{k=1}^{p-1}\frac{1}{n+1+4sin^{2}\left(\frac{k\pi }{p}\right)}+\sum_{k=1}^{p-1}\frac{1}{n+3+4sin^{2}\left(\frac{k\pi }{p}\right)}). \end{array}$$

One can see that the limitation properties for $H(G_{3})$ are the same as those of $H(G_{2})$, i.e., $\lim_{n\to \infty }H\left(G_{2}\right)=\lim_{n\to \infty }H\left(G_{3}\right)$. In addition, the case that $G_{3}$ adheres to star-shaped topologies is similar to Case 2 and is omitted here.

**Remark** **3.**
*In the subcase (i) of Case 1 and Case 2, it can be derived that $\lim_{p\to \infty }H\left(G_{2}\right)<\lim_{p\to \infty }H\left(G_{1}\right)$, which means $H(G_{2})$ behaves better than that of $G_{1}$, and the result is in accord with the edge version of the interlace theorem on Laplacian eigenvalues ([29]). However, deleting edges may reduce the cost of the network, although the operation reduces the robustness of the consensus models. Therefore, accurate calculation of a robustness-related index is necessary for judging whether this trade-off is bearable.*


### 3.2. Coherence for the Special Cases: Triplex Star/ Wheel/ Fan Graph

This subsection mainly studies the special cases of the above triplex structures. In fact, if the hub node set of the partial complete structure has only one node, then the topology $S_{p}\times P_{3}$ can be interpreted as a special case of $G_{1}\left(n,p\right)$ with a star structure in each layer, and $F_{p}\times P_{3}$ can be interpreted as a special case of $G_{2}\left(n,p\right)$ with n=1 (see Figure 5). In addition, $W_{n}\times P_{3}$ can be viewed as a special case of $G_{3}\left(n,p\right)$ with n=1.

Denote the network that owns topology $S_{p}\times P_{3}$ by $T_{1}$, and denote the network that owns $F_{p}\times P_{3}$ by $T_{2}$, that is, it has a fan structure in each layer. Denote $W_{n}\times P_{3}$ by $T_{3}$. Then, the following derivation can be acquired: the Laplacian spectrum of the star structure with *p* vertices has the form

$SL\left(S_{p}\right)=\left(\begin{array}{ccc} 0 & p & 1\\ 1 & 1 & p-2 \end{array}\right)$. It can be obtained that
$$\begin{array}{c} H\left(T_{1}\right)=\frac{1}{6p}\left(\left(p-1\right)+\frac{\left(p-2\right)}{2}+\frac{1}{3}+\frac{p-2}{4}+\frac{1}{p}+\frac{1}{p+1}+\frac{1}{p+3}\right) \end{array}$$

Thus, $H\left(T_{1}\right)\to \frac{7}{24}$ as p→∞.

Similarly,

The coherence of $T_{2}$ can be derived as
$$\begin{array}{cc} H(T_{2})= & \frac{1}{6p}(1+\frac{1}{3}+\frac{1}{p}+\frac{1}{p+1}+\frac{1}{p+3}+\sum_{k=1}^{p-2}\frac{1}{1+4sin^{2}\left(\frac{k\pi }{2(p-1)}\right)}+\sum_{k=1}^{p-2}\frac{1}{2+4sin^{2}\left(\frac{k\pi }{2(p-1)}\right)}\\ \; & +\sum_{k=1}^{p-2}\frac{1}{4+4sin^{2}\left(\frac{k\pi }{2(p-1)}\right)}) \end{array}$$

Therefore, $H\left(T_{2}\right)\to \frac{\sqrt{5}}{30}+\frac{\sqrt{3}}{36}+\frac{\sqrt{2}}{48}\approx 0.152$ as p→∞.

Similarly, it can be derived that $H\left(T_{3}\right)=H\left(T_{2}\right)\to 0.152$ as p→∞, and it is found that $\lim_{p\to \infty }H\left(T_{2}\right)=\lim_{p\to \infty }H\left(T_{3}\right)<\lim_{p\to \infty }H\left(T_{1}\right)$.

**Remark** **4.**
*Similar to the triplex cases, duplex structures can also be discussed and the coherence can be derived similarly. The problems may be generalised into multilayered ones or weighted ones in future research.*


## 4. Simulation

This section mainly presents the comparisons and simulation analysis of the coherence of the three-layered networks. One can see from Section 3 that the algebraic connectivity has the relation: $\lambda_{2}\left(G_{i}\left(n,p\right)\right)=\lambda_{2}\left(T_{k}\right)=1$, i=1,2,3;k=1,2,3, which means the consensus speed of the triplex network is irrelevant to the number of nodes *n* or *p*. It is also found that $\lambda_{2}\left(J_{1}\right)=\lambda_{2}\left(J_{2}\right)=\frac{\left(r+2\right)-\sqrt{{\left(r+2\right)}^{2}-4}}{2}$.

For the network coherence for the triplex networks, the change of $H(G_{1}\left(n,p\right))$ with the parameters is shown in Figure 6. It can be seen that when *p* is fixed, $H(G_{1})\to \infty$ as n→∞; when *n* is fixed and p→∞, the asymptotic properties is relevant to *n*. The point (93,97,0.005002) satisfies the result (i) on $H(G_{1})$.

The variance of the first-order coherence for $G_{2}$ with respect to the parameters *n* and *p* (n,p∈[3,100]) is shown in Figure 7. One can see that when the number of hub nodes *n* is assigned to a relatively small value, the coherence changing trend with *p* is slower than the case that *p* is fixed to a small value and *n* varies. When the values of *n* and *p* are both relatively small, we can see that the robustness of $G_{2}$ is worse than in the case that at least one of *n* or *p* is large enough.

When *n* is a relatively small value (see Figure 7, for instance, n=5,p=132, $H(G_{2})\approx 0.08745$), it is consistent with the result that Section 3.1 presents; i.e., when n=3, $H(G_{2}\left(3,p\right))\to 0.0873$ as p→∞ (Figure 8). Figure 9 describes the change of coherence of $H(G_{2}\left(n,3\right))$, which is also consistent with the fact that Section 3.1 presents; i.e., $H(G_{2}\left(n,p\right))\to 0$ when the value of *p* is fixed. Figure 10 shows that when $\lim_{p\to \infty }H\left(J_{2}\right)>\lim_{p\to \infty }H\left(J_{1}\right)$, the points (198, 0.4603) and (196, 0.2535) satisfy the analysis derivation in Case 2. Figure 11 shows that $\lim_{n\to \infty }H\left(J_{1}\right)=\lim_{n\to \infty }H\left(J_{2}\right)=0.25$. Figure 12 implies the limitation trend of the subcase (iii); that is, when r→∞, $H\left(J_{2}\right)>H\left(J_{1}\right)$, it is consist with the analysis that Remark 2 describes.

In Figure 13, the change of first-order coherence of the triplex star network, i.e., $H(T_{1})$, is described. The value of the point (195, 0.2898) in the figure is consistent with the convergence trend that $H\left(T_{1}\right)\to \frac{7}{24}$. Figure 14 characterises the variance of $T_{i},\;i=2,3$, and we can see that the value of the point (297, 0.1525) satisfies the result of calculations of the coherence limitation that $\lim_{n\to \infty }H\left(T_{i}\right)=\frac{\sqrt{5}}{30}+\frac{\sqrt{3}}{36}+\frac{\sqrt{2}}{48}\approx 0.152$.

## 5. Conclusions

This article mainly studies the robustness-related index of the consensus for the partial complete graph-based triplex MASs, where the robustness can be measure by the first-order network coherence. The graph spectra approach is applied to analyse the system topologies and derive the coherence. For the novel three-layered networks, the relations based on the asymptotic behaviours are determined through the calculations of the coherence. Finally, the coherence of the triplex networks with non-isomorphic graphs are analysed and simulated, and it is found that, based on the condition that the number of nodes in the partial complete substructures, i.e., *n*, tends to infinity, the coherence asymptotic behaviour of the two sorts of non-isomorphic triplex networks will increased by $\frac{r-1}{2(r+1)}$, which is irrelevant to the peripheral vertices parameter *p*. We also find that when *p* tends to infinity, the adding star copies operation will reverse the original size relationship of the coherence under the consideration of the triplex networks.

## Figures and Tables

**Figure 1 entropy-24-01296-f001:**
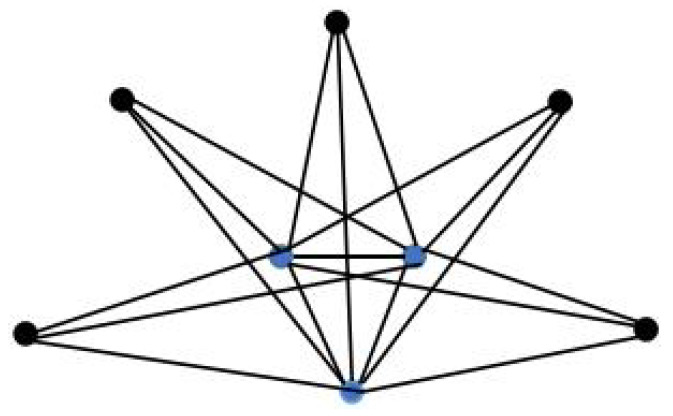
An example of one layer of $G_{1}\left(n,p\right)$ with n=3 and p=5.

**Figure 3 entropy-24-01296-f003:**
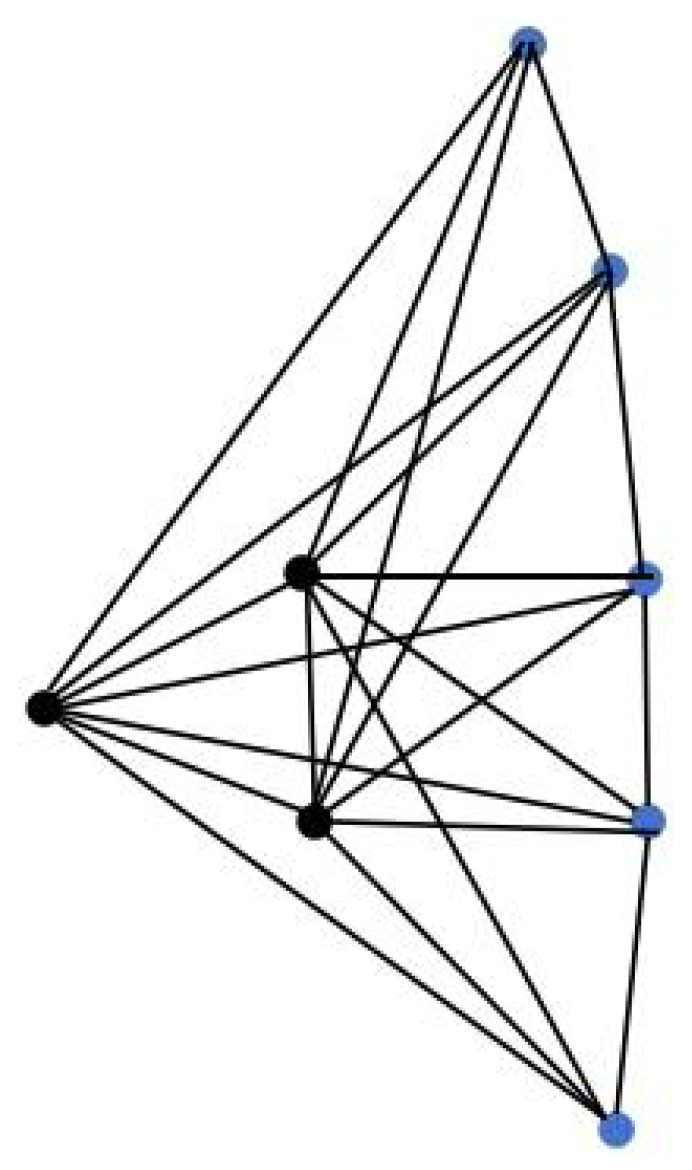
An example of one layer of $G_{2}\left(n,p\right)$ with *n* = 3, *p* = 5.

**Figure 5 entropy-24-01296-f005:**
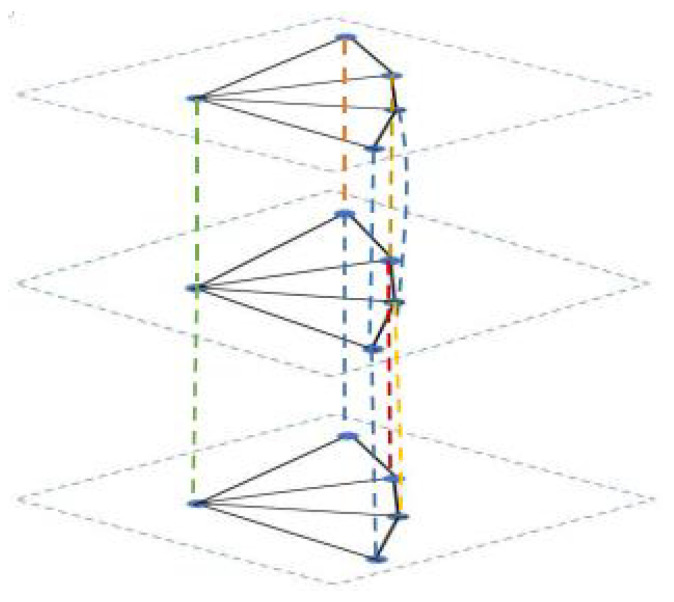
An example of the triplex-fan structure with p=5.

**Figure 6 entropy-24-01296-f006:**
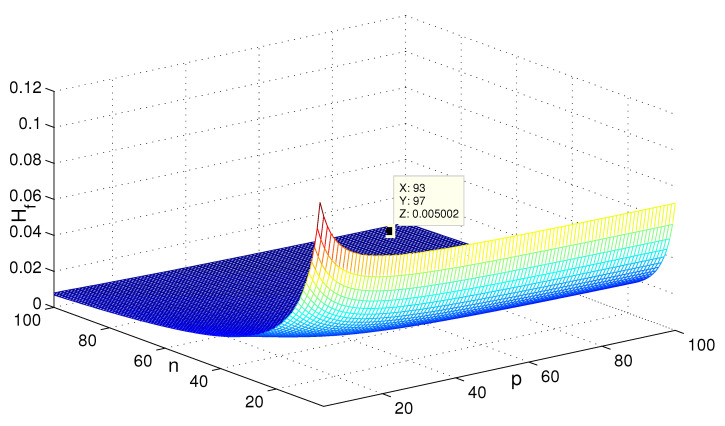
The change of $H(G_{1})$ with the variation of parameters *n* and *p*.

**Figure 7 entropy-24-01296-f007:**
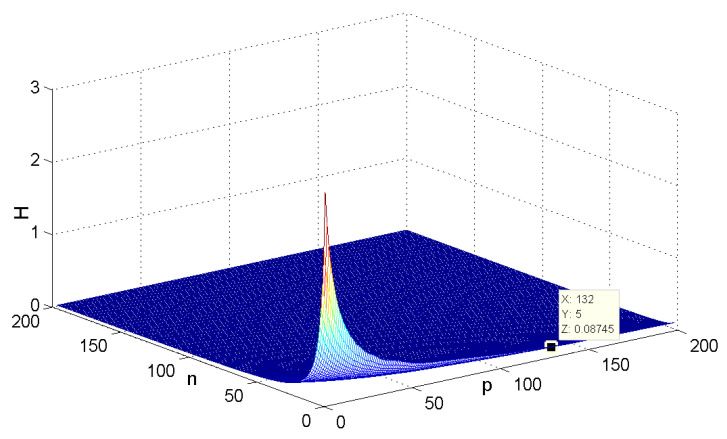
The change of $H(G_{2})$ with the variation of parameters *n* and *p*.

**Figure 8 entropy-24-01296-f008:**
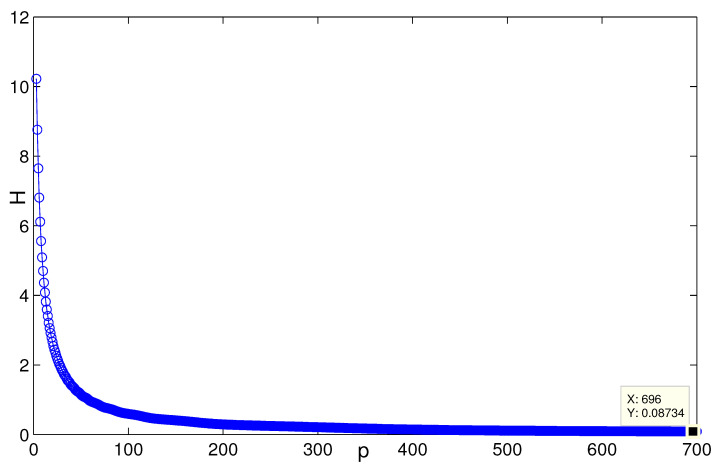
The change of $H(G_{2})$ with *p* when n=3.

**Figure 9 entropy-24-01296-f009:**
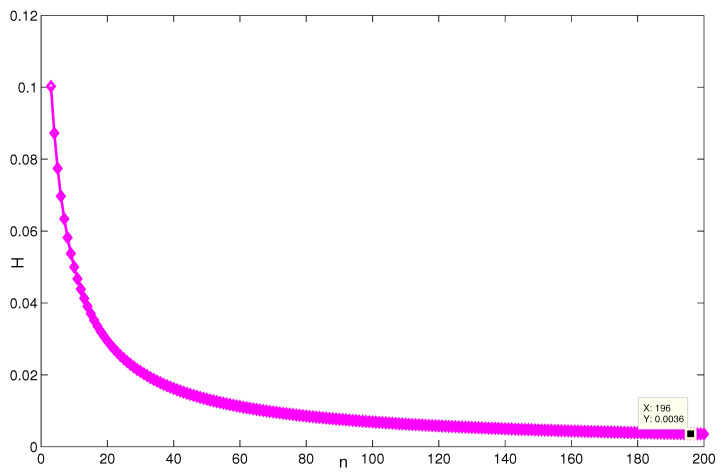
The change of $H(G_{2})$ with *n* when p=3.

**Figure 10 entropy-24-01296-f010:**
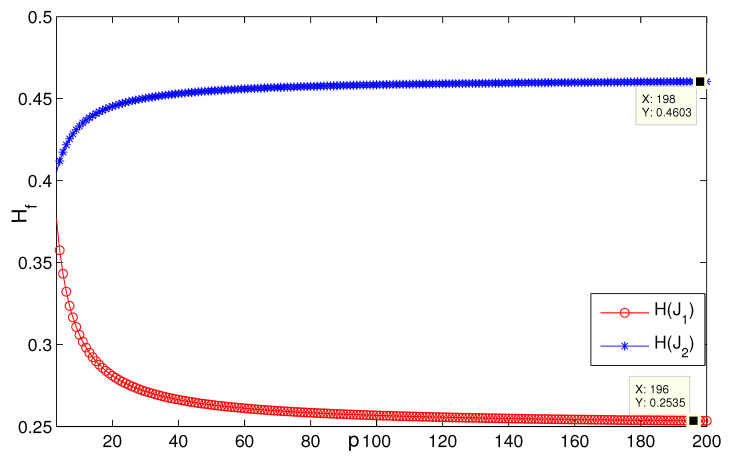
The change of $H(J_{i})$, i=1,2 in the subcase (i) when n=3,r=3.

**Figure 11 entropy-24-01296-f011:**
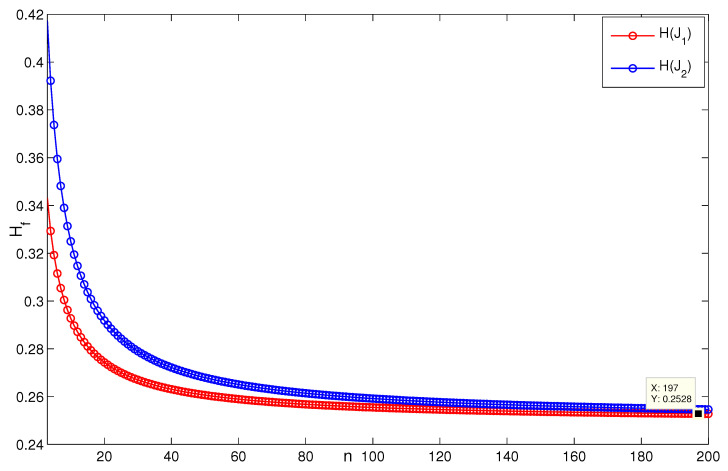
The change of $H(J_{i})$, i=1,2 in the subcase (ii) when r=3,p=5.

**Figure 12 entropy-24-01296-f012:**
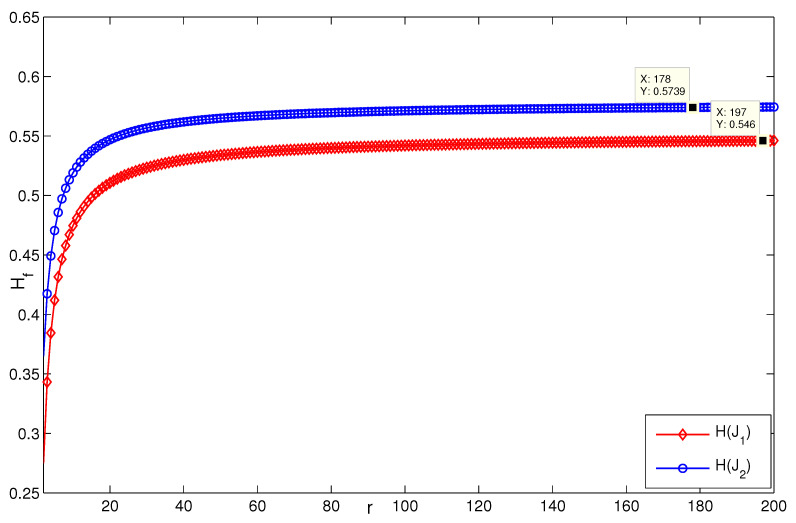
The change of $H(J_{i})$, i=1,2 in the subcase (iii) when n=3,p=5.

**Figure 13 entropy-24-01296-f013:**
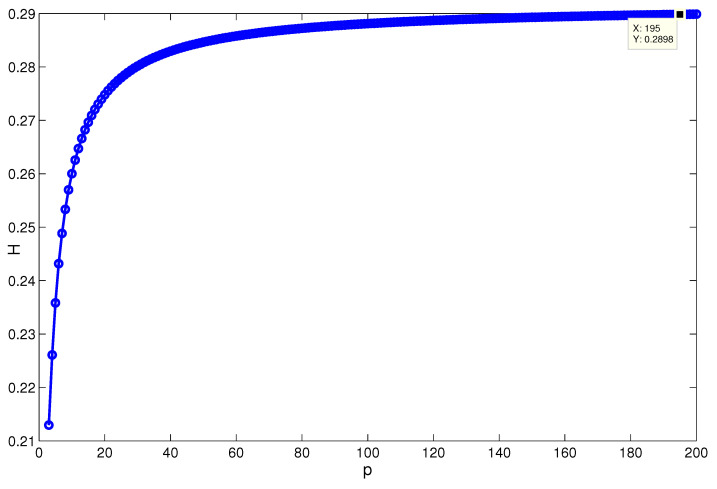
The change of $H(T_{1})$ with respect to *n*.

**Figure 14 entropy-24-01296-f014:**
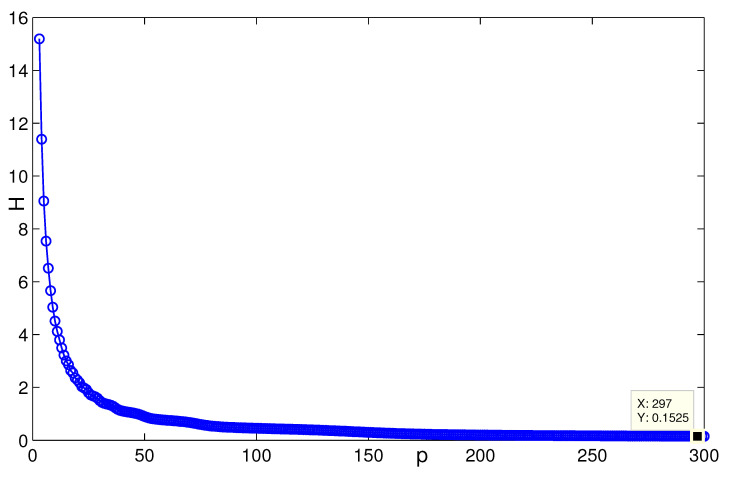
The change of $H(T_{i})$ with respect to *n*, i=2,3.

## Data Availability

Not applicable.

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
