# Peer review of "Consensus-Related Performance of Triplex MASs Based on Partial Complete Graph Structure"

_entropy, 2022, doi:10.3390/e24091296_

Round 1

Reviewer 1 Report

-explain the multiplicity use  (line:176,177,178, ..187)  and its role to compute  the coherence    line:176, 177,178

-lemma 2  is needed     line 209

Author Response

Dear professor,

Thanks very much for your review, and thanks very much for the good valuable advises. We have revised the paper according to the comments and improved the quality of the article. Please see the revised highlighted version.

Reviewer 2 Report

The paper mainly works on the robustness-related index of the consensus for the novel partial complete triplex MASs, where the robustness can be measured by the first-order network coherence. The idea of the work is novel and interesting. However, the quality of the paper is not good enough and I would like to reject the paper as the following concerns.

*Notations are mass.

*The equations are confusing and without any explanations.

*Lines 139 to 148 are not clear.

*Lines 154 to 155 are not clear.

*Lines 176 to 186 are not clear.

*The equation in Line 230 is missed.

 *The presentation should be improved.

Overall, the paper is not good enough. 

Author Response

Dear Professor,

  Thanks very much for your good valuable comments, they are very helpful to improve the quality of the article. please see the attachments for details of reponse and revised article

Reviewer 3 Report

The authors in this paper examined the first-order coherence related to the robustness of the triplex MASs consensus models with partial complete graph structures. They studied the performance index through algebraic graph theory. They generated the topologies of the new triplex networks by different graph operations. They utilized spectral graph theory's concepts to calculate the first-order network coherence. They also compared and analyzed the coherence asymptotic behaviors of three cases of the partial complete structures. In this reviewer's opinion, the obtained results are new and correct, and thus I recommend them for publication. The authors are suggested to remove grammatical errors. Also, I have an optional suggestion: it would be nice if the authors add some additional recent references on the topic.

Author Response

Dear professor,

Thanks very much for your review, and thanks very much for your appreciation. We have revised the paper according to the good valuable advises, the typos and syntax problem has been corrected. Some unnecessary description or sentences have been deleted. Some verbal descriptions have been modified to improve the quality of the paper.(please see the highlighted part of the latest version of paper)

Round 2

Reviewer 2 Report

Thanks for the revisions. 

The current version is better than the previous one. However, the authors did not seem to add some explanations for the presented formula as my suggestion.

It is difficult to follow the work without some explanations. So I would like to see the modifications on explanations.

Author Response

Dear Professor,

  Thanks very much for your good valuable comments, they are very helpful to improve the quality of the article.  We have revised the paper again according to your valuable advise, please see the attachments of response and the previous highlighted version and the revised version of the article.
